# Assessing Suitable Techniques for Rainwater Harvesting Using Analytical Hierarchy Process (AHP) Methods and GIS Techniques

Ammar Adham [1,2] , Michel Riksen [1,*] , Rasha Abed [1,3], Sameer Shadeed [4] and Coen Ritsema [1]

1 Soil Physics and Land Management Group, Wageningen University, 6700 AA Wageningen, The Netherlands; engammar2000@uoanbar.edu.iq (A.A.); rashahameed@mtu.edu.iq (R.A.); coen.ritsema@wur.nl (C.R.)

2 Dams and Water Resources Engineering Department, College of Engineering, University of Anbar, Baghdad 55431, Iraq

3 Anbar Technical Institute, Middle Technical University, Baghdad 10074, Iraq

4 Water and Environmental Studies Institute, An-Najah National University, Nablus 62451, Palestine; sshadeed@najah.edu

* Correspondence: michel.riksen@wur.nl

**Abstract:** The objective of this study is to produce suitability maps for potential rainwater harvesting techniques (RWHT) in the West Bank (WB), Palestine. These techniques aim to reduce water scarcity, which is a major problem for the conservation of water resources in the area. Based on literature reviews and expert recommendations, seven RWHts were selected (runoff basin system, contour ridges, cisterns, eyebrow terrace, check dam, on-farm pond, and bench terraces). Analysis methods performed in the Arc GIS environment include spatial analysis and data reclassification. Other calculations include multi-criteria analysis for assigning suitability. Five criteria (rainfall, runoff, land use, slope, and soil texture) for RWHt were analyzed to produce a suitability map for each technique. The results show that runoff basin systems in the northeast and southwest of WB are the most suitable, with about 50% of the area of WB moderately suitable for this technique, while 70% of the area of WB is very suitable for the contour ridge technique. Furthermore, this analysis shows that almost 50% of the WB is very suitable for cisterns. Sixty percent of the area is very suitable for on-farm puddling, especially in the north and southwest of WB. The areas with high suitability for the different techniques comprehensively cover the WB, as shown in the RWHt suitability maps and the integrated map. Nevertheless, this approach can help decision makers in making an initial selection of RWH techniques suitable for their region.

**Keywords:** rainwater harvesting technique (RWHt); the West Bank (Palestine); analytical hierarchy process method (AHP); GIS

## 1. Introduction

Irregular rainfall patterns and a lack of precipitation have caused water shortages around the world. People living in many areas with highly variable rainfall and unpredictable periods of drought or flooding are severely affected by water scarcity and often face livelihood insecurity [1]. Those regions, including Palestine, are characterized by arid to semi-arid climatic conditions and have uncertain water supplies. Population growth to approximately 2.9 million [2] and expansion of agriculture activities increase the stress on limited and uncertain water supplies; furthermore, the current political situation poses another accessibility limitation of water resources for Palestinians. The water shortage issues include the domestic and the agriculture sector. For domestic water, Palestinian Water Authority (PWA) statistics showed that, in most of the West Bank (WB) governorates, the average water consumption rate is (72 L/capita/d), which lies below the minimum World Health Organization's standards (150 L/capita/d) [3]. According to the Palestinian

Central Bureau of Statistics (PCBS), the total domestic water supply in WB increased from (85 MCM/year) in 2010 to (120 MCM/year) in 2015 [2]. A recent study showed that the water supply–demand gap in the entire WB will increase from (31.7 MCM/year) in 2015 to (41.2 MCM/year) in 2032 [4].

The main agricultural water source in WB is the groundwater (springs and wells) and the share of water purchased from Mekorot (Israeli Water Company, Israel). In 2015, the WB governorates' agricultural water requirement for all crops was (75 MCM/year) with a water supply–demand gap of (46.5 MCM/year) with almost the same amount in 2032 [4]. As a result, alternative water resources such as rainwater harvesting (RWH) are becoming a common practice in most regions of WB [5].

In 2011, rainwater harvesting techniques (RWHt) contribute about 1.5 MCM/year for agricultural use, whereas cisterns contribute to domestic use at about 4 MCM/year [2]. According to the 2018 PWA water plan, 10 MCM/year may be gathered via the use of various domestic and agricultural RWH approaches [6].

As a result, non-conventional water resources, such as RWH, may be used to alleviate water shortages in the WB, Palestine. The implementation of RWHt is promoted on a small scale by local societies and non-governmental establishments to improve temporal and spatial water shortage for domestic and agricultural uses. The success of RWH systems depends heavily on their technical design and the identification of suitable sites and techniques [6,7].

The identification of appropriate sites for the various RWHt in large areas was a great challenge [8]. Several methodologies have been established for the identification of RWH suitable sites. Some methodologies integrate multi-criteria decision making (MCDM) based on geoinformation and SWAT (Soil and Water Assessment Tool) model [9], while others used the TOPSIS multi-criteria decision analysis [10]. An intensive research effort focused on the development of rainwater harvesting (RWH) site suitability maps in different areas [11–13]. In contrast, less attention has been paid to the development of RWHt suitability maps. Most studies rely on the analysis of site characteristics to determine suitability rather than on the analysis of technical characteristics. Therefore, the preparation of RWHt suitability maps is crucial for determining which RWH technology is suitable for each suitable site. Thus, to successfully plan and implement RWHt, it is important to determine suitability for both the site and the technique.

In their literature study, Ammar et al. [14] evaluated primary criteria that have been used to identify potential RWH locations and procedures in arid and semi-arid areas (ASARs). They classified and contrasted four primary site selection methodologies, indicated three main sets of criteria for choosing RWH sites, and defined the most prevalent RWHt utilized in ASARs. Others used different methods [15], ranging from those based only on biophysical factors to more comprehensive ones, which included the inclusion of socioeconomic criteria, particularly after 2000. Most studies currently employ GIS in conjunction with hydrological models and/or multi-criteria analysis (MCA) to identify RHW suitability sites. Shadeed and Alawna [16] used this method to identify locations for the successful implementation of RWHt for agricultural use in the WB, Palestine. They concluded that 62% of the WB is high to very high suitable for implementing RWHt. However, they did not make a distinction between the different available RWHts. There are several differences among these RWHt from functional, construction, and design requirements. Not all techniques are equally suitable for every location. To select the best RWHt, it is better to make a suitability map for each technique separately.

This study aimed at developing suitability maps for potential RWHt in the WB by employing a GIS-based MCA approach. Moreover, this study aims at preparing an integrated RWHt for all the selected techniques that are of high value for water decision makers to properly identify suitable techniques that can be implemented in the area. This in turn will enhance sustainable water resources in Palestine.

## 2. Materials and Methods

### 2.1. Study Area

The West Bank (WB), Palestine, is located in the Middle East (Figure 1) with an area of about 5860 km². It has a population of 2.9 million people distributed in 11 administrative governorates [2].

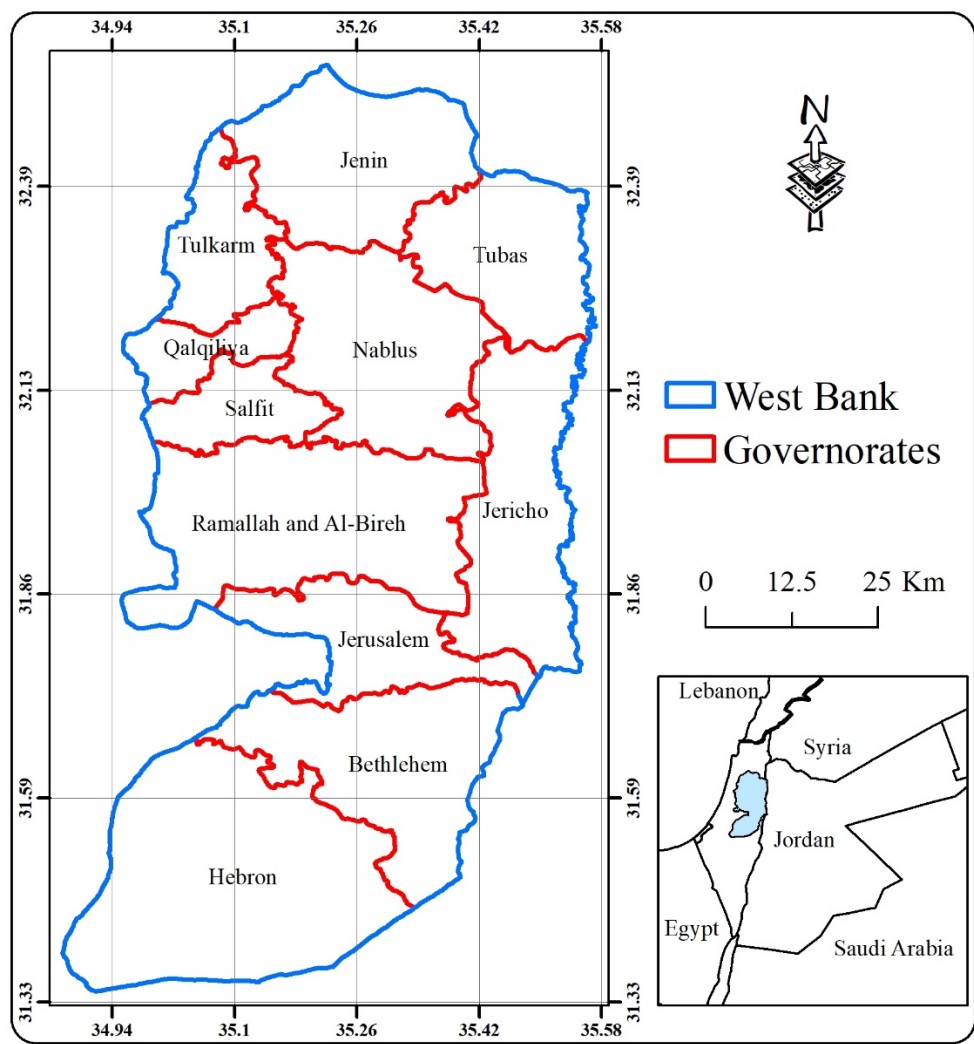

**Figure 1.** Administrative governates of the West Bank, Palestine.

Geographically, the WB is largely made up of hills (700 and 900 m above sea level) that run north-south and then fall to the Jordan Valley and the Dead Sea on the east side.

In general, the predominant climate is the Mediterranean, with rainy winters and hot, dry summers; the eastern and southern parts are much drier [16]. Surface water is mainly in the Jordan River and ephemeral wadis. However, since 1967, Palestinians do not have the right to access the river; therefore, they mainly rely on groundwater, the discharge from the different springs, and water purchased for domestic and agricultural use.

The average rainfall in the region is about 450 (mm/year). However, the majority of the yearly rainfall (about 80%) falls during the winter [16] with an average runoff curve number of about 50 [17], which indicates the potential for implementing RWH.

Widely variable land-use patterns are determined by the accessibility of water. Comparatively well-watered non-irrigated land within the hills is used for the grazing of sheep and tree crops. Irrigated land within the hills and also the river basin is intensively cultivated for various fruits and vegetables [18].

*2.2. Methodology Overview*

The identification of rainwater harvesting techniques (RWHt) suitability maps involved four stages:

i.    Selection of RWHt;
ii.   Selection of appropriate criteria for each technique;
iii.  Suitability classification for each criterion;
iv.   GIS application and maps suitability development.

2.2.1. RWHt Selecting

When developing a rainwater harvesting (RWH) system, selecting the appropriate RWHt after determining the RWH site that meets the fundamental technical design requirements for rainwater harvesting is a critical component for assuring long-term implementation. There are numerous strategies and approaches that have been employed to conserve rainwater all over the world.

The design requirements for RWHt as well as the assessment of their site-suitability play a major role in determining technique suitability. The first step in the selection process was to decide which rainwater harvesting techniques could be mapped at the country level, as RWH is a site-specific activity. First, data were collected with respect to the most commonly practiced RWHt in districts with similar climatic conditions and topography.

Next, an overview of the critical values for climate, soil conditions, topography, and other variables affecting each RWHt was created and a pre-selection list for the RWHt was made.

Finally, the pre-selected list was discussed with local experts, taking into account the already implemented RWHt. As a result, seven RWHts were selected: runoff basin system, contour ridges, cisterns, eyebrow terrace, check dam, on-farm pond and bench terraces.

2.2.2. Criteria Selection

This step formulates the set of criteria to choose suitable RWHts based on the primary purpose, expert assessments, literature studies, and, most significantly, data availability. To classify RWHt suitability and influenced by the RWH site suitability criteria, five criteria were selected: annual precipitation (rainfall) as a climate parameter, runoff and curve number (CN) as a hydrology parameter, land use as an agronomy parameter, slope as a topography parameter, and soil texture as a soil parameter.

I.    Rainfall

In any RWH system, the amount and distribution of rainfall are essential factors in determining whether a specific RWHt is suitable or not in a particular location. In ASARs, rainfall is characterized by high temporal and spatial variation [19]. When designing rainwater harvesting systems, the catchment region should receive enough rainfall for storage for future use.

II.   Runoff depth (curve number, CN)

The depth of runoff is used to determine the amount of water available during runoff. The runoff depth was calculated using the curve number (CN) given by the Soil Conservation Service [20]. The effects of soil and land cover on rainfall and runoff predict CN. Land-cover and soil-texture maps were used to estimate CN for each pixel in the research region. The depth of runoff may be stated as follows:

$$Q = \frac{(P - I_a)^2}{(P - I_a) + S} \tag{1}$$

where $Q$ = depth of runoff (mm), $P$ = precipitation (mm), $S$ = potential maximum retention (mm), and $I_a$ = initial abstraction (mm).

$I_a$ = 0.2 S based on the analysis of rainfall data from small agricultural basins [21]. As a result, Equation (1) may be written as follows.

$$Q = \frac{(P - 0.2S)^2}{(P + 0.8S)} \tag{2}$$

*S* may be worked out using CN as follows.

$$S = \frac{25400}{\text{CN}} - 254 \tag{3}$$

The runoff reaction to a given rain is represented by CN, which ranges from 0 to 100. The presence of high CNs indicates that a significant percentage of the rainfall will be the surface runoff [22,23]. Shadeed and Almasri [17] created the CN map for the entire WB.

III.　　Land use (LU)

At a given location, the runoff generated by rainfall is related to the use of the land. For example, when the land is more densely vegetated, there is less surface runoff and there will be more water infiltration [24]. In the WB and depending on the Ministry of Agriculture (MoA) database, there are seven different types of land uses that have been discovered: built-up, woodland, grazing, irrigated farming, permanent crops, Arab land, and Israeli settlements.

IV.　　Slope

Slope plays a direct role in selecting the appropriate RWHt since it interferes with the structure's design. On another hand, it has a significant impact on the runoff generation and, thus, on the sedimentation amount, water flow velocity, and the cost (in terms of time, materials, and effort) required to implement RWH [25]. In Arc GIS 10.2, a 30 m resolution digital elevation model (DEM) was utilized to create the slope map.

V.　　Soil texture

In RWHt design, soil texture affects both surface runoff and soil infiltration rates [26,27]. The percentage of sand, silt, and clay in a soil determines its texture class. According to Adham et al. [1], soil with fine and medium texture is usually better suited for RWH as they retain water better. Soil texture is certainly one of the most important factors in deciding where to build an RWHt. This importance varies from one technique to another depending on the design and operation of each technique. In this study, four soil texture classes were selected Sandy loam, Loamy, Clay loam, and Clay [28].

### 2.2.3. Criterion Suitability Classification

In this step, the different values within the different datasets were converted into a common suitability scale using GIS and MCA. Due to the different measures and weights for the different criteria, each of the five criteria was first classified. The wight for each criterion was determined after assigning scores using the Analytic Hierarchy Process (AHP) and the pairwise comparison matrix [29]. When comparing and scoring two criteria, a continuous 9-point scale is used, with odd numbers 1, 3, 5, 7, and 9 representing the range of suitability (not suitable–very suitable) of the criteria compared to each other. The scores were assigned and adjusted based on active discussions with local experts and engineers, as well as on information from previous scientific work.

### 2.2.4. GIS Application and Maps Suitability Development

In creating the datasets, the DEM and rainfall station data required additional analysis to derive the input data maps for slope and precipitation. Based on the suitability scale, a new scaled map was created for each input layer. The Spatial Analyst module of Arc GIS 10.2 was used to determine suitability by reclassifying the criteria layers and using the raster calculation tool. The final step is to combine the converted output layers of

annual perception, land use, slope, soil texture, and CN. For each RWHt, each criterion was classified as a numerical value and assigned a suitability value. Then, the suitability values were classified into four groups: low (<20), moderate (20–29), high (30–39), and very high (>39). Figure 2 shows the flowchart of the steps taken to derive the RWH suitability map for each RWHt. A Model Builder in ArcGIS10.2.1 was established for generating a suitability model, which included steps for calculating the suitability score for each RWHt.

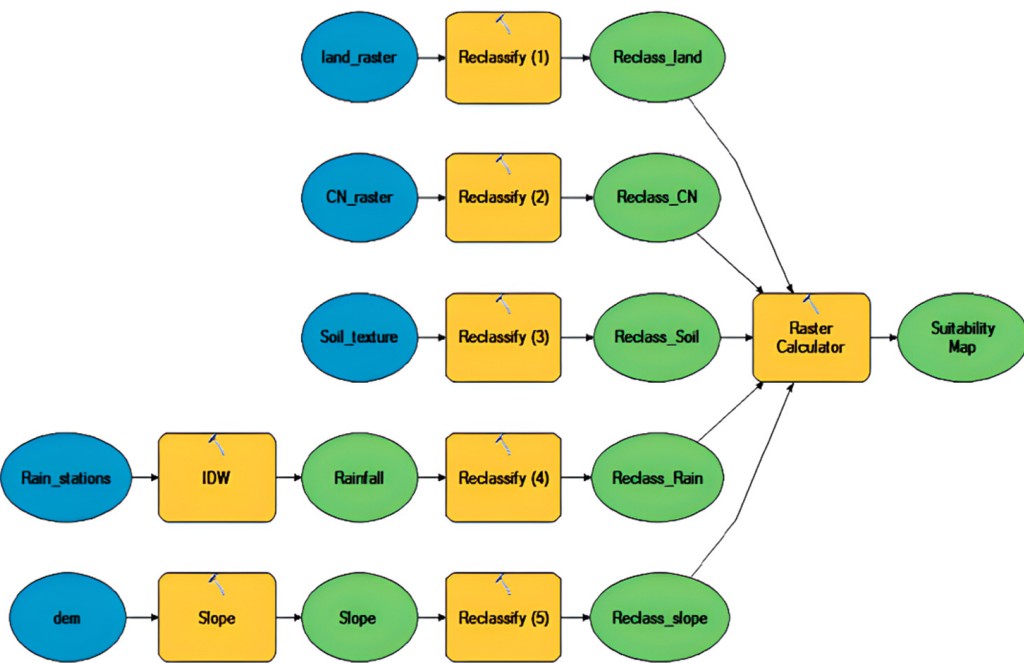

**Figure 2.** Flow chart for the identification of RWHt suitability map.

## 3. Results and Discussion

### 3.1. Input Maps

Figure 3 shows the GIS input maps representing the levels in our model for the suitability analysis. The long-term average annual precipitation for the entire West Bank (WB) is shown in Figure 3a. The amount of precipitation varies greatly within the WB. In particular, the eastern and southern parts of the WB are much drier. Potential runoff, shown in the CN figure in Figure 3b, is low in areas with sandy loam soil in Figure 3f and very high in built-up areas in Figure 3c. Permanent crops, including arable crops, and irrigated farming are found mainly in the regions with higher rainfall. Pasture dominates in the eastern part of the WB. Figure 3d shows the slope driven from the dem in Figure 3e.

### 3.2. Suitability Score for Each Criterion for the Rwhtt

All seven selected rainwater harvesting techniques (RWHts) were assigned a suitability scale. All layers were reclassified according to their suitability with a specific score, as shown in Table 1. The suitability ratings and criteria selection were the result of several discussions with local experts and engineers with experience in developing RWHt. The ratings were updated and modified several times depending on the previous studies to avoid discrepancies in the allocation of points [25,30–32].

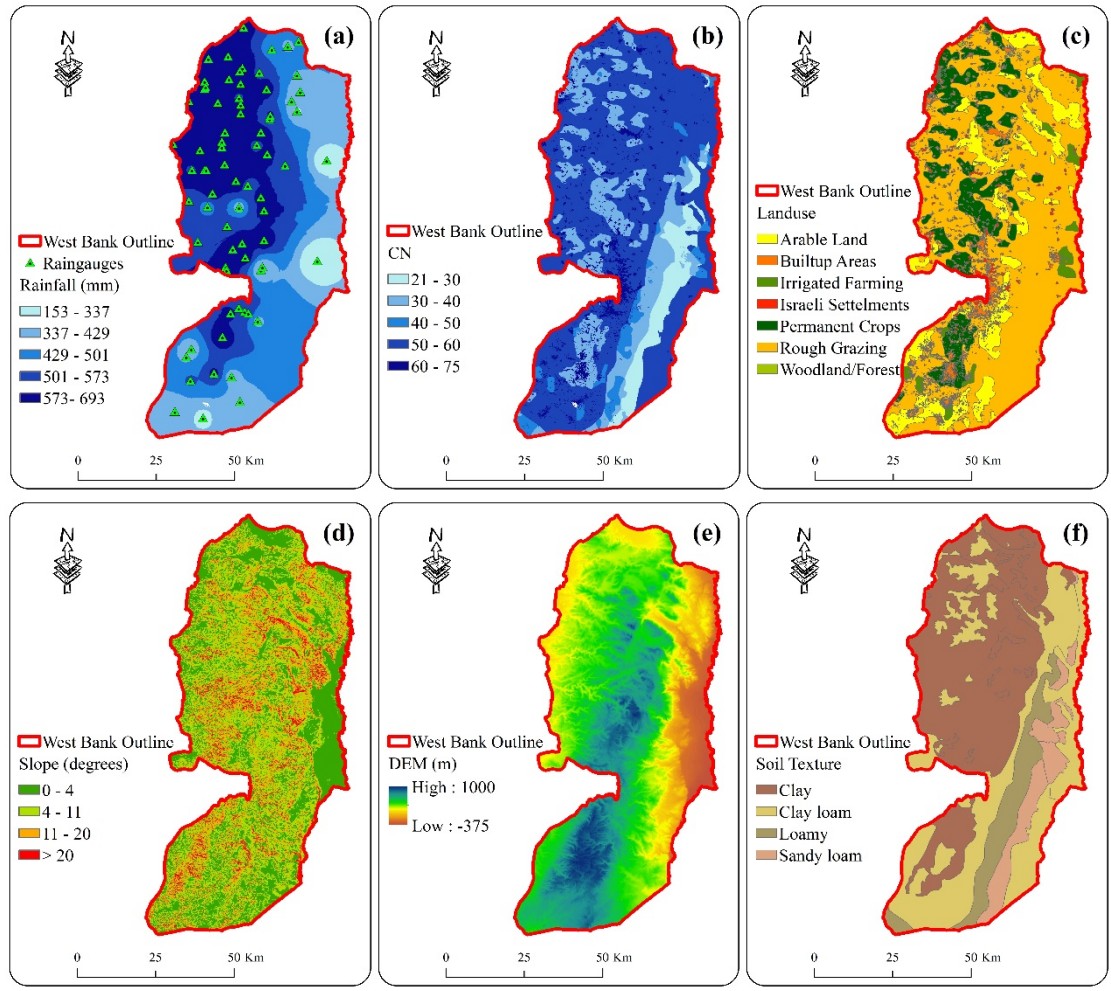

**Figure 3.** Input maps for the WB: (**a**) annual average rainfall (**b**), runoff curve number, (**c**) land use, (**d**) slope, (**e**) dem, and (**f**) soil texture.

**Table 1.** The suitability score of the selected rainwater harvesting techniques for the West Bank, Palestine.

| # | Criteria | Classes | Score | | | | | | |
|---|----------|---------|-------|---|---|---|---|---|---|
| | | | Runoff Basin | Contour Ridges | Cistern | Eyebrow Terrace | Check Dam | On-Farm Pond | Bench Terrace |
| 1 | Annual rainfall (mm) | <250 | 7 | 3 | 9 | 7 | 5 | 5 | 5 |
| | | 250–500 | 9 | 9 | 7 | 9 | 7 | 7 | 9 |
| | | 500–750 | 3 | 7 | 5 | 5 | 9 | 9 | 7 |
| 2 | Land use | Arable Land (supporting grains) | 7 | 9 | 1 | 7 | 5 | 7 | 7 |
| | | Built-up Areas | 1 | 1 | 5 | 1 | 1 | 1 | 1 |
| | | Woodland/Forest | 1 | 1 | 1 | 3 | 1 | 1 | 1 |
| | | Rough Grazing/Subsistence Farming | 3 | 5 | 1 | 5 | 5 | 5 | 3 |
| | | Irrigated Farming | 1 | 3 | 9 | 1 | 9 | 9 | 1 |
| | | Permanent Crops (Fruits trees) | 9 | 7 | 7 | 9 | 3 | 3 | 9 |
| | | Israeli Settlements | 1 | 1 | 1 | 1 | 1 | 1 | 1 |

**Table 1.** *Cont.*

| # | Criteria | Classes | Runoff Basin | Contour Ridges | Cistern | Eyebrow Terrace | Check Dam | On-Farm Pond | Bench Terrace |
|---|---|---|---|---|---|---|---|---|---|
| | | | Score | | | | | | |
| 3 | Slope (%) | flat (0–2) | 9 | 3 | 3 | 7 | 5 | 3 | 1 |
| | | gentle (2–5) | 7 | 5 | 9 | 9 | 9 | 7 | 1 |
| | | moderate (5–10) | 5 | 9 | 7 | 5 | 7 | 9 | 3 |
| | | rolling (10–15) | 1 | 7 | 5 | 3 | 3 | 5 | 7 |
| | | hilly (15–30) | 1 | 1 | 1 | 1 | 1 | 1 | 9 |
| | | steep >30 | 1 | 1 | 1 | 1 | 1 | 1 | 5 |
| 4 | Soil texture | Sandy loam | 3 | 5 | 1 | 5 | 3 | 3 | 7 |
| | | Loamy | 5 | 7 | 3 | 7 | 5 | 5 | 5 |
| | | Clay loam | 9 | 9 | 5 | 9 | 7 | 7 | 9 |
| | | Clay | 7 | 3 | 9 | 3 | 9 | 9 | 3 |
| 5 | Curve number | ≤50 | 3 | 5 | 3 | 3 | 3 | 3 | 5 |
| | | 51–60 | 7 | 7 | 5 | 7 | 5 | 5 | 7 |
| | | 61–70 | 9 | 9 | 7 | 9 | 7 | 7 | 9 |
| | | >70 | 5 | 3 | 9 | 5 | 9 | 9 | 3 |

### 3.3. The Potential of RWHt

Figure 4 shows potential maps for different types of RWHt. The maps identified by the spatial analyst module show the suitability on a scale from red (not suitable) via yellow (moderate suitable) to green (very high suitable), based on the five selected criteria. Table 2 shows the percentage per suitability class for each RWH technique for the entire WB.

**Table 2.** Percentage of the WB suitable for different RWH techniques.

| Suitability Score | Low | Moderate | High | Very High |
|---|---|---|---|---|
| | <20.0 | 20 to 29 | 30 to 39 | >40 |
| On farm pond | 3.4 | 52.9 | 42.8 | 1.0 |
| Bench terraces | 0.9 | 29.6 | 69.5 | 0.0 |
| Check dam | 1.7 | 40.1 | 51.7 | 6.4 |
| Eyebrow terraces | 14.8 | 72.8 | 12.3 | 0.0 |
| Cistern | 0.0 | 53.7 | 46.2 | 0.1 |
| Contour ridges | 3.3 | 33.0 | 59.9 | 3.8 |
| Runoff basin | 6.3 | 64.4 | 28.7 | 0.5 |

To obtain an overview of the suitability assessment for all selected techniques together, Figure 5 shows an integrated suitability map created with specific analysis properties in Arc GIS 10.2. Each RWHt has been assigned a specific color to indicate its high suitability for a specific area in WB. The map shows a large discrepancy in the amount of land suitable for the different RWHts.

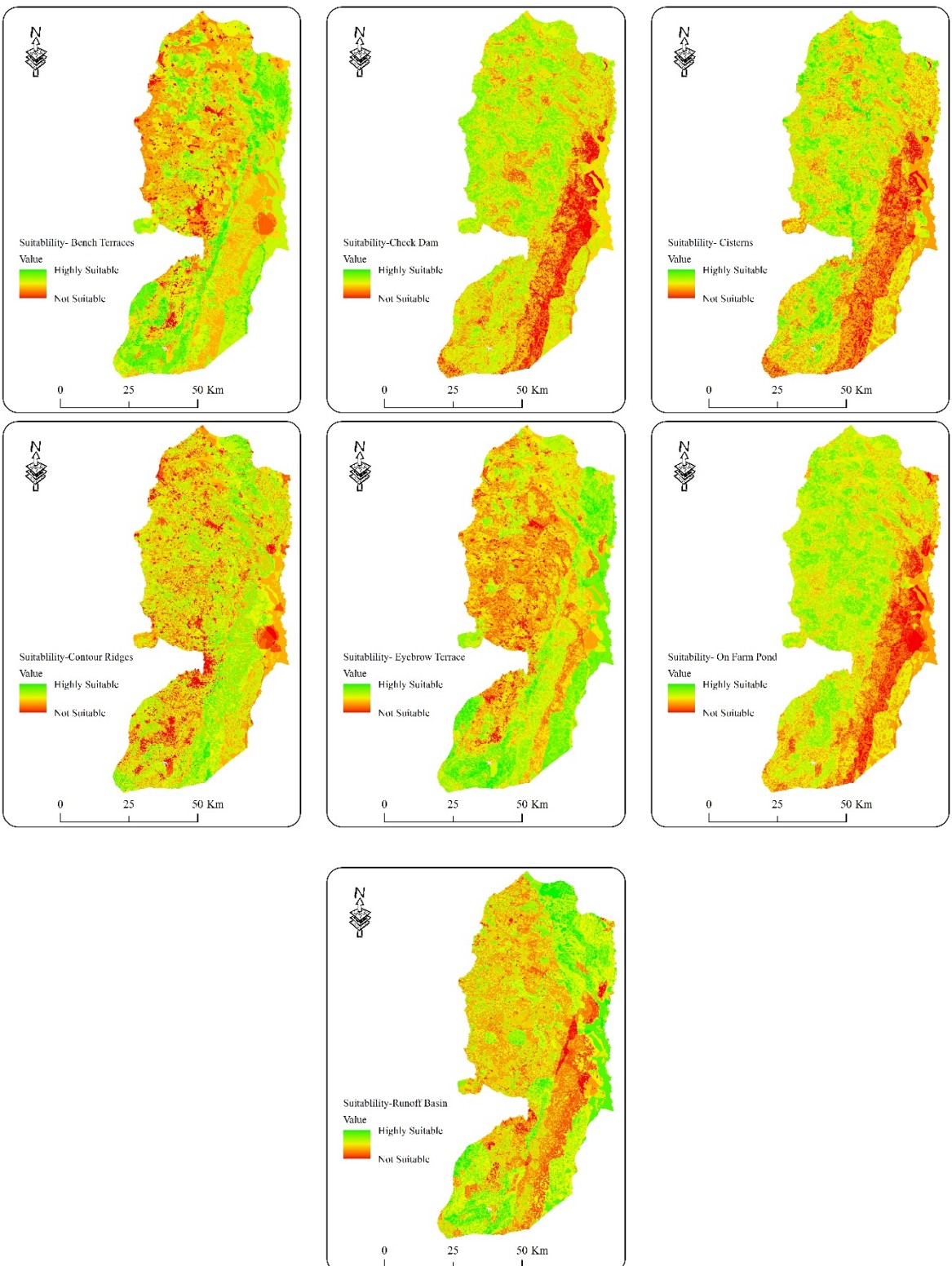

**Figure 4.** Suitability maps for different types of RWHt in the WB, Palestine, based on soil type, slope, rainfall, land use, and curved number.

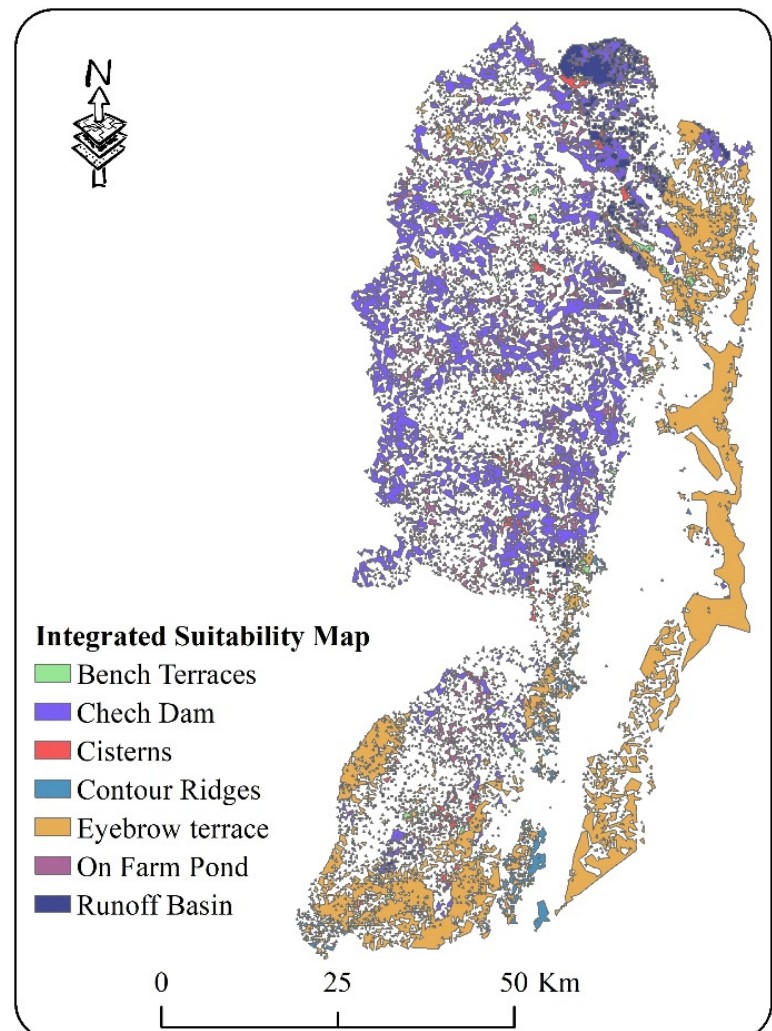

**Figure 5.** Integrated suitability map of all the seven RWHts for the WB, Palestine.

From the maps in Figure 4 and the integrated map in Figure 5, it is clear that the different RWHts have different suitability in WB. Each RWHt has its own suitability map, which reflects the technical requirements for that technique and is influenced by the criterion. Runoff basins are most suitable on the northern eastern and southern western borders of WB in the relatively shallow parts, and the soil is mostly clay loam. Map statistics show that about 50% of the area of WB is moderately suitable for this technique. Seventy percent of the area of WB is very suitable for the contour ridge technique. Most of the well-suited sites are in the northern east and southern east with cropland and a perceptual range of more than 250 mm. About 50% of the total area of WB is very suitable for the cistern technique. These areas are mainly in the northwest and southwest where the soil texture is mainly classified as clay loam, as the infiltration rate in the catchment area of the cistern is an important factor for suitability. Therefore, the area with high infiltration rate and low CN value shows low suitability for the cistern.

Highly suitable areas for implementing eyebrow terraces are located mainly on the northern and southeastern borders and in the southwestern borders of WB. Statistics showed that about 70% of the area of WB is moderately suitable for this technique. This can be explained by the fact that this technique is less suitable for clay soils. As for the farm pond, the areas with high suitability for this technique are mainly located in the north and southwest of WB, with 60% of the area with moderate slopes and clay soil being very suitable for this technique.

It should be noted that according to this analysis, about 80% of the area of WB, which is mainly in the north and northwest, is highly suitable for the construction of check dams. Figure 3d shows that these areas have a slope of 15 to 30% and a CN of 60 to 70 mm. Areas with high suitability for the use of the bench terrace technique are located in the northeast and southwest of WB where orchard farms are located and these areas have a hilly to steep slopes. Map statistics show that more than 50% of the area of WB is highly suitable for this technique.

A look at the input maps in Figure 3, the scoring system in Table 1, and the result of the individual RWH suitability map in Figure 4 shows how the suitability maps are strongly influenced by the score assigned to each criterion.

The relatively high suitability scores for these RWH techniques do not mean that these are the best solutions for farmers, as socioeconomic, political, and cultural aspects were not considered in this analysis. Bench terraces, for example, are too expensive in most cases due to high labour costs. Eyebrow terraces are less suitable for mechanised cropping operations because of their irregular shape. Nevertheless, this approach can help farmers and decision makers in the initial selection of RWHts suitable for their region.

## 4. Conclusions

A suitability model based on GIS, created with ModelBuilder in Arc GIS 10.2, was used to identify potential RWHts. A set of criteria (rainfall, runoff, slope, land use and soil texture) were included in the suitability model.

According to the results of this study, this research technique provides an initial meaningful screening of broad areas and is an extremely useful tool for assisting in the development and implementation of a rainwater harvesting (RWH) project, especially in arid and semi-arid environments. Arc GIS 10.2 has proven to be an extremely useful tool in this study for integrating various information to identify ideal locations for different RW. In screening vast regions for the applicability of RWH measures, Arc GIS 10.2 proved to be a versatile, time-saving and cost-effective tool.

Hydrologists, decision makers, and planners will benefit from the suitability map as it will allow them to easily identify which rainwater harvesting technique (RWHt) to use in sites with RWH potential. The quality and accuracy of the data, as well as the way the data were sourced, processed, and produced, were all factors in the quality of the map.

However, to confirm the applicability of the model, it needs to be calibrated and tested in different regions and with different RWHts. In addition, as the suitability ratings have a major impact on the maps of RWHt suitability, a validation study or pilot project is recommended to ensure the margin of error (if any) in determining the preferences for each RWHt. Socio-economic criteria such as investment and maintenance costs and labour input may also be important for water harvesting. Therefore, socio-economic suitability for different RWHts needs to be explored and included in the assessment process. These ideas will improve the realism of the model and broaden the scope of this methodology.

**Author Contributions:** Conceptualization A.A., M.R. and R.A.; methodology, A.A. and R.A.; software, M.R. and R.A.; formal analysis, S.S. and M.R.; investigation, A.A. and S.S.; resources, A.A. and C.R.; writing—original draft preparation, A.A. and R.A.; writing—review and editing, C.R.; visualization, A.A., S.S. and M.R.; supervision, C.R.; project administration, C.R. and M.R.; funding acquisition, C.R. All authors have read and agreed to the published version of the manuscript.

**Funding:** This Research is fully funded by NUFIC Orange Knowledge Program, 9OKP-IRA-104278.

**Data Availability Statement:** Some data in this manuscript were obtained from the Ministry of Agriculture and PWA, Palestine. The other data are from the fieldwork and previous studies.

**Conflicts of Interest:** The authors declare no conflict of interest.

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
