# Peer review of "Assessing Suitable Techniques for Rainwater Harvesting Using Analytical Hierarchy Process (AHP) Methods and GIS Techniques"

_water, doi:10.3390/w14132110_

Round 1

Reviewer 1 Report

General comments

The identification of suitable sites for RWH is an important step towards maximizing water availability and land productivity in Arid and Semi-Arid regions (ASARs). In reality, the main purpose of this study is to define suitable sites for different RWH techniques applications. It is therefore dealing with a matter of particular and international interest.

In 1995, it was established the Palestinian Water Authority (PWA) which aims to achieve integrated and sustainable asset management of ed water resources, protection, and preservation, and the ensuring a balance between quantity and quality of water available and the needs of the Palestinian people to achieve sustainable management of the water resources. In my opinion, a session should be added briefly describing the measures taken, and the policy followed by PWA. I consider it would help the international reader to understand the problem better and make the article probably more attractive.

In the study area (West Bank, Palestine), various studies have been done using almost the similar tools for the same purpose (suitable RWH site selection) as the manuscript. Authors should report what other scientists have done in the study area, what are their findings, which must comment on them. They should also comment on the differences and similarities between these studies in comparison with their findings. (e.g. (1) Shadeed, S. Developing a GIS-based Suitability Map for Rainwater Harvesting in the West Bank, Palestine. In Proceedings of the International Conference on Professional Environmental Education for Sustainable Development: Plugging the Hole, Birzeit University, Palestine, November 16 – 17, 2011, 13 pp., 2011. Available at https://staff.najah.edu/en/publications/7339/ (accessed on June 03, 2022); (2) Shadeed, S.; Almasri, M. Application of GIS-based SCS-CN method in WB catchments, Palestine. Water Sci Eng 2010, 3(1), 1-13)

However, the manuscript is not written well. It requires major effort to be publishable.

Specifically:

The manuscript has to be written according to the instructions for Authors of the WATER Journal.

A general check of the references is needed (References list).

References have to be written in accordance with the Instructions for Authors of the journal.

a.         Pay attention to the Journal Name, which should be in italic and abbreviated form (see instructions for authors) and the year in bold.

b.         According to the “Reference List and Citations: Style Guide for MDPI Journals”:

For every author, list the last name first, then the first letter of the first name and, if available, the first letter of the middle name.

Last names should be separated from first and middle name by a comma, first and middle names should end with a period: Lastname, F.M.; Lastname, F.M.; etc.

Grammar needs to be remedied at various places of the manuscript. The final text has to be prepared with more care. It includes linguistic errors, that sometimes make it difficult to follow. I would respectfully recommend that Authors have their manuscript professionally edited before resubmitting it or proofreading it with a colleague whose native language is English. Special attention should be paid to the verb tenses.

 Punctuations need to be revised.

 The abstract should be shortened to a total of about 200 words maximum in a single paragraph. It must place the issue addressed in a broad context and highlight the purpose of the study. It must, also, (a) describe briefly the main methods applied; (b) summarize the main findings of the study, and (c) mention the main conclusions. The abstract should be an objective representation of the article. It must not contain results which are not presented and substantiated in the main text, and should not exaggerate the main conclusions.

 Variables in the main text, Tables and in the displayed equations should be in italic.

The subsection entitled “2.2. Methodology overview" (Lines 114 - 223) must be re-written carefully, clearly in correct English.

Figures 1, 3d, 3e and 5 do not mention in the main text.

Table 1 referred to the main text for the first time very far (Line 333) from its appearance in the main text (Line 247).

Table 1 is not understood. More information is required. The reference to the Αnnex complicates rather than simplifies matters.

Table 2 does not mention in the main text.

Figure 4 referred to the main text for the first time very far (Line 334) from its appearance in the main text (Line 301).

The section entitled “3. Results and Discussion” (Lines 224 – 341) is very unclear, not understood. It must be re-written carefully, clearly in correct English. According to the Instructions for Authors of the Journal, "Authors should discuss the results and how they can be interpreted in perspective of previous studies and of the working hypotheses. The findings and their implications should be discussed in the broadest context possible and limitations of the work highlighted. Future research directions may also be mentioned. This section may be combined with Results".

The section entitled “4. Conclusions” (Lines 342 – 366) must be re-written carefully, clearly in correct English. It would be useful to indicate sources of uncertainties and limits of applicability of the evidences emerged in the study. In addition, it is required a brief explanation of the significance and implications of the work.

Annex 1: The definition and clarification of each criterion is not clear. Authors should provide much more information about each criterion so that the reader understands how it is evaluated for each technique.

 Specific comments

Lines 17 – 19

The Authors state: “Five biophysical criteria have been analysed (rainfall, runoff, land use, slope, and soil texture) for RWH techniques to produce suitability map for each technique”.

COMMENT: However, the rainfall is a climate factor, the rainfall is a hydrology factor, land use is an agronomy factor, slope is a topography factor and soil texture is a soil factor. How the Authors justify the term “biophysical criteria” and specifically, the first word ("bio") from the compound word "biophysical"

Lines 19 - 27

Results of data analysed show that Runoff basin systems Runoff basin are most suitable in the northern east and southern west borders of the WB around 50% of the WB area is moderately suitable for this technique. While 70% of the WB showed high suitability for the contour ridge technique. Moreover, this analysis shows that almost 50% of the West Bank is highly suitable for the cistern. On- farm pond high suitability areas are mostly located in the north and southern west parts of the WB with 60% of the area is highly suitable. The high suitability areas for the different techniques covered the West Bank area in a relatively comprehensive way as shown in the different RWH techniques suitability maps and in the integrated map. However, this approach can help the farmers to make a first selection of RWH techniques suitable for their region”.

COMMENT: This part of the text is unclear and unintelligible.

Lines 37 - 38

“The water shortage issues included the domestic and the agriculture sector”.

should be

“The water shortage issues include the domestic and the agriculture sector”.

Lines 40 & 41

“litters/capita/day” should be “L/capita/d” or alternatively “Lpcd”

COMMENT: According to the Instructions for Authors of the Journal the International System of Units (SI) should be used. The symbols of “litters” is “L” (capital) and of “day” is “d”. Many times in the international literature the symbol “Lpcd” is used

Lines 42 - 43

“from 85 million cubic meters MCM/year”

should be

“from 85 MCM/year”

COMMENT: Delete “million cubic meters”. No need. Τhe symbol of “million cubic meters” in SI units is “MCM”.

Lines 49 & 52

“… MCM per year …”

should be

“…MCM/year …”

Line 53

“…with about 4 MCM [2]”

should be

“…with about 4 MCM/year [2].”

Line 54

“….water plan, 10 MCM may be gathered…”

should be

“….water plan, 10 MCM/year may be gathered…”

Line 62

“…in large areas was a great…”

should be

“…in large areas is a great…”

Lines 66 - 68

With the availability of different RWH suitability maps in different regions produce the critical need to identify which RWH technique is suitable for each location.

COMMENT: This sentence is unclear.

Lines 69 – 74

Throughout the past three decades, Ammar [11] evaluated the methodologies and primary criteria that have been used to identify potential RWH locations and procedures in arid and semi-arid areas (ASARs). They classified and contrasted four primary site selection methodologies, indicated three main sets of criteria for choosing RWH sites, and defined the most prevalent RWHt utilised in ASARs. Methods ranged from those based only on biophysical factors to more comprehensive ones, which included the inclusion of socioeconomic criteria, particularly after 2000”.

COMMENT: This part of the main text is unclear and erroneous. Ammar et al [11] (no Ammar [11]) presented a review paper concerning the definition of a general method for selecting suitable RWH sites in ASARs by assembling an inventory of the main methods and criteria developed during the last three decades.

Line 76

“…to identify RHW…”

should be

“…to identify RWH…”

Line 81

“Shadeed [12] used this method…”

Should be

“Shadeed and Alawna [12] used this method …”

Lines 84 - 85

The Authors stated: “There exists, however, a large difference between these RWHt.”

COMMENT: Clarify what does it mean, “large difference between these RWHt”.

Lines 87 – 91

“The aim of this study was to develop suitability maps for potential RWHt in the WB by employing a GIS-based MCA. Moreover, preparing an integrated RWHt for all the selected techniques which are of high value for water decision-makers to properly identify suitable sites for the implementation of different RWHt. This in turn will enhance sustainable water resources in Palestine.”

COMMENT: The aforementioned text is not clear. It should be rewritten carefully, clearly, in correct English.

Lines 94 - 96

The WB, Palestine is located in the Middle East Error! Reference source not found.. It covers about 5860 km2 and a population of 2.9 million people from 11 administrative governorates [2].”

COMMENT: This part of the text is incorrect and not understood.

Line 99

“Geographically, The WB is largely….”

should be

“Geographically, the WB is largely….”

Line 106

“The average rainfall in the region is about 450 (mm) per year.”

should be

“The average rainfall in the region is about 450 mm/year.”

Lines 106 – 108

“However, the majority of the yearly rainfall (about 80%) falls during the winter [12] with an average runoff curve number of about 50 assuming dry conditions [13], which indicate the potential for implementing RWH”.

COMMENT: This sentence is unclear. It must be rewritten clearly.

Lines 124 – 126

“The technical design of the techniques, as well as the assessment of their site-suitability, play a big role in the technique’s selection”.

COMMENT: The above sentence must be re-written in correct English.

Line 159

“As a result, Eq1 may be written as:”

should be

“As a result, Equation (1) may be written as:”

Lines 183 – 186

“This importance varies from Technique to another considering the designe and functioning of the technique. In this study and according to literature recommendation, four soil texture classifications were selected; Sandy loam, Loamy, Clay loam, and Clay.”

COMMENT: The aforementioned text is not clear. It should be rewritten carefully, clearly, in correct English.

Lines  209 - 210

“…shows the flow chart of the steps taken to derive the RWH suitability map for each RWHt.

COMMENT: This part of the text is incorrect and not understood.

Lines  222 – 223

Delete the sentence “Following that, the suitability values were divided into four groups: low (score < 20), moderate (20 – 29), high (30 -39) and very high (> 39)”.

COMMENT: No need. It's a duplication (see Lines 208 – 209).

Lines 231, 232, 233

“Fig.” should be “Figure”

Lines 240 -242

“Error! Reference source not found. shows the suitability scores which were the result of several discussions with local experts and engineers with experience in designing RWH techniques”.

COMMENT: This part of the text is incorrect and not understood.

Lines 250 – 255

“Error! Reference source not found. shows potential sites maps for different types of RWH. The maps show the sites those are identified by the spatial analyst module showing the suitability on a scale from red (not suitable) via yellow (moderate suitable) to green (very high suitable), based on the five selected criteria. Error! Reference source not found. shows the percentage per suitability class for each RWH technique for the whole West Bank”.

COMMENT: This part of the text is incorrect and not understood.

Lines 258 - 262

“In order to have an over view about the suitability score for all the selected techniques together, Error! Reference source not found. shows an integrated suitability map was generated using special analysis properties in ArcGIS. A specific color was given to each RWHt to indicate its high suitability to specific area in the WB. The map shows a large discrepancy in the amount of land suitable for the different RWHt”.

COMMENT: This part of the text is incorrect and not understood.

Lines 305 - 311

From the maps in Error! Reference source not found. and the integrated map in Error! Reference source not found., it becomes clear that the different RWH techniques show different suitability levels in the WB. Each RWH technique suitability map reflect the technical requirements for that technique. Runoff basins are most suitable in the northern east and southern west borders of the WB in the relatively flat parts and the soil is mostly clay loam. The map statistics showed that around 50% of the WB area is moderately suitable for this technique”.

COMMENT: This part of the text is incorrect and not understood.

Line 334

“…in Figure 4 can shows how …”

should be

“…in Figure 4 can show how …”

Line 383

“Int. Soil Water Conserv, (2018), 6(4),”

should be

“Int. Soil Water Conserv 2018, 6(4),”

Line 392

“Journal of water and health, (2011), 9(3), 525-533”

should be

J Water Health 2011, 9(3), 525-533”

Lines 393 – 394

Alawna, S, and S. Shadeed Rooftop Rainwater Harvesting to Alleviate Domestic Water Shortage in the West Bank, Palestine. An - Najah Univ. J. Res. (N. Sc.) (2021), Vol. 35(1),. https://journals.najah.edu/article/1788/

should be

Alawna, S.; Shadeed, S. Rooftop Rainwater Harvesting to Alleviate Domestic Water Shortage in the West Bank, Palestine. An - Najah Univ. J. Res. (N. Sc.) 2021, 35(1), 83-108. Available at: https://journals.najah.edu/article/1788/ (accessed on Day Month Year).

Lines 395 – 396

Adham, A., Riksen, M., Ouessar, M., Abed, R., & Ritsema, C.. Development of Methodology for Existing Rainwater Harvesting Assessment in (semi-) Arid Regions. In Water and Land Security in Drylands, (2017), (pp. 171-184). Springer International Publishing.

should be

Adham, A.; Riksen, M.; Ouessar, M.; Abed, R.; Ritsema, C.  Development of Methodology for Existing Rainwater Harvesting Assessment in (semi-) Arid Regions. In Water and Land Security in Drylands: Response to Climate Change; Ouessar, M.; Gabriels, D.; Tsunekawa, A.; Evett, S., Eds.; Springer International Publishing: Cham, Switzerland, 2017, pp. 171-184; ISBN 978-3-319-54021-4

Line 404

“Sustainable Water Resources Management, (2021), 7(1), ….”

should be

“Sustain Water Resour Manag 2021, 7(1), …..”

Lines 406 – 407

Ammar, A.; Riksen, M.; Ouessar, M.; Ritsema, C. Identification of suitable sites for rainwater harvesting structures in arid and semi-arid regions: A review. Int. Soil Water Conserv. Res. (2016).

should be

Adham, A.; Riksen, M.; Ouessar, M.; Ritsema, C. Identification of suitable sites for rainwater harvesting structures in arid and semi-arid regions: A review. Int Soil Water Conserv Res 2016, 4, 108–120.

Lines 416 – 417

Computers and Electronics in Agriculture, (2002), 37(1–3), 173–183”

should be

Comput Electron Agric 2002, 37(1-3), 173-183”

Line 419

Applied Geography. (2014), 51, pp. 131-142.”

should be

Appl Geogr 2014, 51, 131-142.”

Line 421

Agricultural Water Management, (2016b), 176, 191–202.”

should be

Agric Water Manag 2016, 176, 191 – 202.”

Lines 422 – 423

Kahinda, Lillie, E. S. B., Taigbenu, A. E., Taute, M., & Boroto, R. J.. Developing suitability maps for rainwater harvesting in South Africa. Physics and Chemistry of the Earth, (2008), 33(8–13), 788–799. http://doi.org/10.1016/j.pce.2008.06.047.

should be

Mwenge Kahinda, J.; Lillie, E.S.B.; Taigbenu, A.E.; Taute, M.; Boroto, R.J. Developing suitability maps for rainwater harvesting in South Africa. Phys Chem Earth 2008, 33, 788–799

Lines 424 – 425

Adham, A., Riksen, M., Ouessar, M., & Ritsema, C. J. A Methodology to Assess and Evaluate Rainwater Harvesting Techniques in (Semi-) Arid Regions. Water, (2016a) 8(5), 198.

should be

Adham, A.; Riksen, M.; Ouessar, M.; Ritsema, C.J. A Methodology to Assess and Evaluate Rainwater Harvesting Techniques in (Semi-) Arid Regions. Water 2016, 8, 198; doi:10.3390/w8050198.

Lines 426 – 427

Journal of Environmental Assessment Policy and Management, (2008), 10(2), 189–206”

should be

J Environ Assess Policy Manag 2008, 10(2), 189-206.”

Lines 428 – 429

Al-Adamat, R., AlAyyash, S., Al-Amoush, H., Al-Meshan, O., Rawajfih, Z., Shdeifat, A., Al-Farajat, M.. The combination of indigenous knowledge and geo-informatics for water harvesting siting in the Jordanian Badia. (2012)

should be

Al-Adamat, R.; AlAyyash, S.; Al-Amoush, H.; Al-Meshan, O.; Rawajfih, Z.; Shdeifat, A.; Al-Harahsheh, A.; Al-Farajat, M. The Combination of Indigenous Knowledge and Geo-Informatics for Water Harvesting Siting in the Jordanian Badia. J Geogr Inf Syst 2012, 4, 366-376.

Lines 431 - 433

Umugwaneza, A., Chen, X., Liu, T., Mind'je, R., Uwineza, A., Kayumba, P. M., ... & Maniraho, A. P. Integrating a GIS-based approach and a SWAT model to identify potential suitable sites for rainwater harvesting in Rwanda. AQUA—Water Infrastructure, Ecosystems and Society, (2022), 71(3), 415-432.

should be

Umugwaneza, A.; Chen, X.; Liu, T.; Mind’je, R.; Uwineza, A.; Kayumba, P.M., Uwamahoro, S.; Umuhoza, J.; Gasirabo, A.; Maniraho, A.P. Integrating a GIS-based approach and a SWAT model to identify potential suitable sites for rainwater harvesting in Rwanda. AQUA — Water Infrastructure, Ecosystems and Society 2022, 71(3), 415 – 432. doi: 10.2166/aqua.2022.111

Line 433

Saaty, T. L. Decision making with the analytic hierarchy process. Int. J. Serv. Sci. (2008), 1, 83.

should be

Saaty, T.L. Decision making with the analytic hierarchy process. Int J Serv Sci 2008, 1(1), 83 – 98.

Line 435

“Annexes” should be “Annexe”

Annex 1 (Lines 436 – 437)

Line 1 Column 3: Consider if you can use as header the term "description" instead of the term "definition".

Line 2 Column 4: The Authors state: “High score is given to the optimum design requirements (Literature)”

COMMENT: Which literature are you referring to? Add citation.

Line 3 Column 4: The Authors state: “For this criterion, the scores were given based on literature recommendations, the design requirements of the techniques, and local experts recommendations.”

COMMENT: Which literature recommendations are you referring to? Add citation.

Author Response

Please find our response in the attached file 

Reviewer 2 Report

Dear authors,

Please check the attached annotated file. You will find all my comments there. Check sections and sub-sections and respond point by point.

Thanks and best of luck

Author Response

(The authors gave the same response as above.)

Round 2

Reviewer 1 Report

1.         The Authors made effort to improve the quality of the manuscript. However, it is still not well written. There are errors, ambiguities, omissions, etc., that must be corrected before its publication. Consequently, the manuscript needs major try to become publishable. For details please, see my comments.

2.         The manuscript has to be written according to the Instructions for Authors of the WATER Journal.

3.         Grammar needs to be remedied at various places of the manuscript. The final text has to be prepared with more care. It includes linguistic errors, that sometimes make it difficult to follow. I would respectfully recommend that Authors have their manuscript professionally edited before resubmitting it or proofreading it with a colleague whose native language is English.

4.         Punctuations need to be revised.

5.         The whole manuscript must be read carefully and corrected. Many parts of the main text are unclear and unintelligible. In my comments, I highlighted only a few of them. I considered that my attempt to highlight all the points in the main text was futile and excessive.

6.         Lines 2 – 3

Title:

“Assessing sites suitable for Rainwater Harvesting Techniques, using GIS based AHP Method”

should be

“Assessing suitable sites for Rainwater Harvesting Techniques, using Analytical Hierarchy Process Method (AHP) and GIS technique”.

7.         Line 29

Keywords: The West Bank; Palestine; GIS; rainwater harvesting; suitability map

should be

Keywords: Rainwater Harvesting Technique (RWHt), The West Bank (Palestine), Analytical Hierarchy Process Method (AHP), GIS

8.         Define the abbreviations WB, RWH, RWHt

COMMENT: According to the Instructions for Authors of the Journal the abbreviations should be defined in parentheses the first time they appear in the abstract, main text, and in figure or table captions and used consistently thereafter. In the present case, the abbreviation RWH appears first time in Line 13 and its definition becomes on Line 52. Moreover, the term rainwater harvesting techniques appear on Line 16, and its abbreviation becomes on Line 54.

9.         In the references list, there are 30 citations in total from which the five (5) of them (1-7-14-24-23) belong to the first, second, and fifth co-author of the manuscript and the four of them (4-6-15-16) belong to the fourth. Consequently, there is a disproportionate share (about 1/3) of the citations which belong to the co-authors of the manuscript. The authors have to take note of this highlighting and correct it.

10.     Line 99

“…in the Middle East Figure 1…”

should be

“…in the Middle East (Figure 1)…”

11.     Line 100

“…about 5860 km2 and has a population…”

should be

“…about 5860 km2. It has a population…”

12.     Line 104

“Geographically, The WB is largely…”

should be

“Geographically, the WB is largely…”

13.     Line 122

Clarify the term “basic technical design requirements for harvesting rainwater”. What exactly does this term mean?

14.     Lines 127 – 129

“When developing a RWH system, selecting the appropriate approaches after determining the area that meets the fundamental technical design requirements for rainwater harvesting is a critical component for assuring long-term implementation.”

COMMENT: This sentence is unclear. Clarify the term “appropriate approaches after determining the area”. What exactly does this term mean?

15.     Lines 132 - 133

“The technical requirements for the design of the techniquesRWHt as well as the assessment of their site-suitability play a major role in the selection of the technique.””

COMMENT: This sentence must be written in correct English

16.     Lines 138 – 142

“Next, an overview of the critical values for climate, soil conditions, topography and other variables affecting each RWHt was made and a pre-selection was made. Finally, the pre-selected list was discussed with local experts, taking into account the already implemented RWHt. As a result, seven RWHt were selected: Runoff Basin System, Contour Ridges, Cisterns, Eyebrow Terrace, Check Dam, On-Farm Pond and Bench Terraces.”

COMMENT: This part of the main text is unclear and unintelligible

17.     Lines 146 - 147

“To classify RWHt suitability, and influenced by RWH site suitability criteria, five biophysical criteria were selected;”

should be

“To classify RWHt suitability, and influenced by RWH site suitability criteria, five criteria were selected;”

COMMENT: Please change the term “biophysical criteria” in the main text in “criteria”. The term “biophysical” involves biological and physical factors. In the present case, the rainfall is a climate factor, the runoff is a hydrology factor, land use is an agronomy factor, slope is a topography factor, and soil texture is a soil factor.

18.     Lines 158 – 159

“…given by the Soil Conservation Service (Add citation)”

19.     Lines 3, 18, 29, 31, 184, 210, 216, 224, 231, 262, 347, 353, 355)

The authors mention elsewhere “Arc GIS 10.2” (Lines 184, 210, 224,347), elsewhere “ArcGIS10.2.1” (Line 216), elsewhere “ArcGIS” (Lines 18, 262, 353, 355) and elsewhere “GIS” ( Lines 3, 29, 231, 347).

COMMENT: Keep the same term throughout the main text.

20.     Line 196

Clarify the term “suitability scale”. What exactly does this term mean?

21.     Line 240

“…for The WB: (a) annual….”

should be

“…for the WB: (a) annual….”

22.     Lines 244 – 247

“The suitability ratings and criteria selection were the result of several discussions with local experts and engineers with experience in developing RWHt. The ratings were updated and modified several times depending on the previous studies to avoid discrepancies in the allocation of points. [24,29,30]”

COMMENT: The above text is very unclear. According to the Instructions for Authors of the Journal, the Authors should be described (materials and methods) with sufficient detail to allow others to replicate and build on published results. Consequently, more information and details need concerning the suitability rating and the selection of criteria.

23.     Line 253

“Figure 4Error! Reference source not found. shows potential”

COMMENT: This part of the main text is not understood.

24.     Lines 394 – 395 & 415 - 416

The citations No 6 and No 15 are the same? Check, please.

“6. Alawna, S, and S. Shadeed. Rooftop Rainwater Harvesting to Alleviate Domestic Water Shortage in the West Bank, Palestine. An - Najah Univ. J. Res. (N. Sc.) 2021, 35. https://journals.najah.edu/article/1788/”

“15. Alawna, S.; Shadeed, S. Rooftop Rainwater Harvesting to Alleviate Domestic Water Shortage in the West Bank, Palestine. An - Najah Univ. J. Res. 2021, 35, 83-108.”

Author Response

Please find our response in the attached file.

Reviewer 2 Report

Thank you for incorporating all the suggestions.

Author Response

We appreciate your effort to give specific comments to modify our manuscript.